# Modelling radiative transfer through ponded first-year Arctic sea ice with a plane parallel model

Torbjørn Taskjelle[1], Stephen R. Hudson[2], Mats A. Granskog[2], and Børge Hamre[1]

[1]Department of Physics and Technology, University of Bergen, Allégaten 55, Bergen, Norway
[2]Norwegian Polar Institute, Fram Centre, Tromsø, Norway

*Correspondence to:* Torbjørn Taskjelle, torbjorn.taskjelle@uib.no

**Abstract.** Under-ice irradiance measurements were done on ponded first-year pack ice along three transects during the ICE12 expedition north of Svalbard. Bulk transmittances ($400$–$900\,\mathrm{nm}$) were found to be on average 0.15–0.20 under bare ice, and 0.39–0.46 under ponded ice. Radiative transfer modelling was done with a plane parallel model. While simulated transmittances deviate significantly from measured transmittances close to the edge of ponds, spatially averaged bulk transmittances agree well. That is, transect-average bulk transmittances, calculated using typical simulated transmittances for ponded and bare ice weighted by the fractional coverage of the two surface types, is in good agreement with the measured values. Radiative heating rates calculated from model output indicates that about 20% of the incident solar energy is absorbed in bare ice, and 50% in ponded ice (35% in pond itself, 15% in the underlying ice). This large difference is due to the highly scattering surface scattering layer (SSL) increasing the albedo of the bare ice.

## 1 Introduction

As summer reaches the Arctic Ocean and temperatures rise, the snow covering the sea ice will melt and form melt ponds. Melt ponds dramatically change the optical properties of the sea ice, as ponds have a much lower albedo than the surrounding bare ice, and transmittance of solar radiation is generally higher through ponds than through adjacent ice that is not covered by ponds (e.g. Perovich et al., 1998). On a larger scale, the timing for the onset of melt, and thereby occurrence of melt ponds, has been shown to possibly influence the yearly sea ice area minimum (Markus et al., 2009; Schröder et al., 2014; Liu et al., 2015), as well as the annual budgets of solar energy (Arndt and Nicolaus, 2014). Melt pond coverage is an important controlling factor of sea ice albedo (Eicken et al., 2004; Perovich, 2005; Divine et al., 2015).

First-year ice (FYI) typically has larger, but shallower, ponds than multi year ice (MYI), due to the latter having a rougher topography in general (Eicken et al., 2004; Webster et al., 2015). Over the last few decades the fraction of FYI in the Arctic has increased, while MYI has declined (e.g. Stroeve et al., 2011).

Simulating the irradiance field passing through ponded ice is, in principle, a problem requiring a 3D radiative transfer model. However, it is often more practical to use a plane-parallel model. Here we show that at least for the specific case at hand, a plane-parallel model can give useful results, if the desired result is a spatial average. Nevertheless, local results at locations close to the boundary between ponded and bare ice will be inaccurate, due to the large contrasts in surface properties. Lu et al.

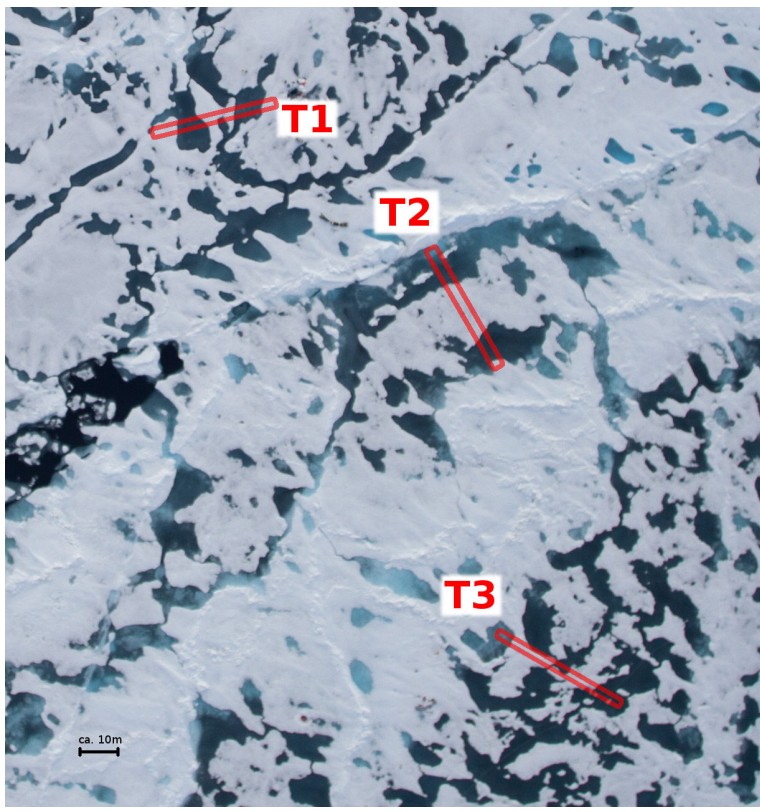

**Figure 1.** Aerial photo of part of the study area taken from a helicopter on 28 July 2012, with the three transects marked. The labels are placed at the start of each transect. The length of the transects 1, 2, and 3, were approximately 33 m, 35 m and 38 m, respectively. The whole image covers approximately $180\,\mathrm{m} \times 200\,\mathrm{m}$.

(2016) investigated the effect of pond depth and ice thickness on albedo and transmittance of ponded sea ice, using a two-stream radiative transfer model, but did not consider edge effects. Neither did Light et al. (2008) or Light et al. (2015) consider edge effects, focusing instead on a few select case studies. Petrich et al. (2012) used a 3D Monte Carlo model to estimate that 90% of the irradiance measured at a point at the ice bottom will be incident in a radius that is twice the ice thickness. In the
study by Ehn et al. (2011) edge effects were seen up to $4.4\,\mathrm{m}$ away from the pond-bare ice–boundary, for ice thicknesses up to $1.77\,\mathrm{m}$. Ponds have also been seen to influence the irradiance field several meters down in the water column below bare ice (Frey et al., 2011; Katlein et al., 2016).

Few studies with high spatial resolution transmittance data across ponds and adjacent bare ice are available. In addition to that by Ehn et al. (2011), Nicolaus et al. (2012) and Nicolaus and Katlein (2013) used data from multiple flights with a remotely
operated vehicle (ROV) under ponded ice, but the former focussed more on the differences between FYI and MYI, while the latter was a more methodological study, discussing the use of ROVs under sea ice. A disadvantage of some ROV studies is that the ROV is operated too far below the ice on horizontal transects to completely observe local effects.

In this study we present data from ponded first-year ice in an advanced stage of melt north of Svalbard, where under-ice transmittance was measured with the help of divers (Hudson et al., 2013). Radiative transfer modelling was performed with a plane parallel radiative transfer model. Model results are used to quantify the radiative heating rate of the ponds and sea ice, and limitations of using a plane parallel model in cases with a highly heterogeneous surface such as ponded sea ice are discussed.

## 2 Methods and data

### 2.1 Observations

Data were collected on first-year ice north of Svalbard during the ICE12 expedition on R/V *Lance* in July–August 2012, between $82.5°$N and $82°$N (Hudson et al., 2013). The ice cover was generally less than $1\,\mathrm{m}$ thick and was at an advanced stage of melt, with a melt pond coverage of 25–30% (Divine et al., 2015). The observations for this study were made on one floe that was studied during an 8-day drift (Hudson et al., 2013). Based on aerial surveys the ice was representative of the area (Divine et al., 2015).

Downward irradiance at the ice-ocean interface was measured at intervals of about $1\,\mathrm{m}$ along three different transects by a diver holding a RAMSES Acc-Vis sensor (TriOS Mess- und Datentechnik GmbH, Rastede, Germany). An aerial photo of the study area, with the three transects indicated, is shown in fig. 1. To guide the diver, a rope was stretched under the ice between two poles mounted at the beginning and end of a transect, at a depth of roughly $1\,\mathrm{m}$ below the ice. The rope was marked every $1\,\mathrm{m}$, but due to the elasticity of the rope, the actual distance between markers was somewhat longer, depending on how tight the rope was. An irradiance spectrum was collected directly at the ice bottom above each marker. In order to estimate the true position of the measurements along the transects, variations in the measured ice topography (fig. 4) was compared with variations in pressure measured by the RAMSES sensor. Rope stretch factors of 1.125, 1.05 and 1.1 were estimated for the transects 1, 2, and 3, respectively (cf. tab. 1).

Incident irradiance was measured coincidentally, with the same type of instrument as below the ice, but mounted on a tripod by the dive hole, near the start of the transect. We assume no horizontal variability of incident irradiance between the measuring site, and the transect. Spectra were collected simultaneously from the two sensors, and as RAMSES sensors are generally not calibrated to exactly the same wavelengths, the spectra were interpolated to a common wavelength grid, with $1\,\mathrm{nm}$ spacing.

Ice cores for determining the vertical structure of the ice, in terms of temperature, salinity and microstructure, were not collected from the transects themselves, but temperature measurements from cores taken from bare ice elsewhere on the same ice floe showed that the ice was warm, generally around $0°$C in the surface, decreasing to about $-1.2°$ at the bottom. Assuming a bulk salinity of 3, and temperatures up to $-0.5°$C for the interior ice, the brine volume fraction may be up to 12–30% (Cox and Weeks, 1982; Leppäranta and Manninen, 1988). No samples were melted and filtered for measurement of absorption by particulate matter and colored dissolved organic matter (CDOM).

Ice thickness and freeboard were measured directly by drilling through the ice at $1\,\mathrm{m}$ intervals, after the irradiance measurements were done. The measured ice thickness was used to identify the measurements that were far enough away from a

different surface type to not be influenced by it. For each boundary between pond and bare ice along a transect, the ice thickness $h_i$ for bare ice that was closest to a pond edge was picked out, and all measurements that were done within a distance of $2h_i$ of the pond edge were identified as being close to a pond edge. The value of $2h_i$ was chosen based on (Petrich et al., 2012), who, using a 3D Monte Carlo radiative transfer model, found that 90% of the light hitting a sensor at the ice bottom was incident 5  within a radius of twice the ice thickness, for uniform ice.

## 2.2 Software

The radiative transfer model used is AccuRT (Hamre et al., 2014) (formerly `c-disort`; Hamre et al., 2013). AccuRT is a 1D plane parallel coupled atmosphere–ice-ocean radiative transfer model based on DISORT (Thomas and Stamnes, 1999), and uses the discrete ordinates method to solve the radiative transfer equation. The model domain consists of two horizontal slabs 10  with different refractive indices, and multiple layers in each slab to resolve vertical variations in inherent optical properties (IOPs). Snow can be represented as a layer of ice spheres in the bottom of the upper slab (atmosphere), and ice can be added to the top of lower slab. Brine pockets are represented by spheres of pure sea water with a given radius and volume fraction, and air pockets are likewise represented by spheres of air. Their inherent optical properties are calculated using a parameterization based on Mie calculations (Stamnes et al., 2011).

AccuRT outputs multiple properties, including downward and upward planar ($E$) and scalar ($E_0$) irradiance at specified wavelengths and vertical levels. For the planar irradiance, both the direct beam and the diffuse part of the radiation field are available. The inherent optical properties of each layer are also available.

Analysis and plotting were done using Python, libraries including `numpy` (van der Walt et al., 2011), `scipy` and `pillow`, with plots made in `matplotlib` (Hunter, 2007). Python code for reading model output from AccuRT and calculating heating 20  rate, albedo and transmittance is available at https://github.com/TorbjornT/pyAccuRT.

## 2.3 Theory

In addition to the upward ($E_{0\uparrow}$) and downward ($E_{0\downarrow}$) scalar irradiance at specified depths, AccuRT outputs the total absorption coefficients ($a$) of each layer. The spectral heating rate at a given depth, $H_\lambda(z)$, is determined by the change in net irradiance (upwelling minus downwelling planar irradiance, $E_{\text{net}} = E_\uparrow - E_\downarrow$) with depth,

$$H_\lambda(z) = \frac{\mathrm{d}E_{\text{net}}}{\mathrm{d}z}. \tag{1}$$

This can be related to the scalar irradiances and absorption coefficients output by the model through Gershun's law (e.g. Mobley, 2014),

$$\frac{\mathrm{d}E}{\mathrm{d}z} = -aE_0, \tag{2}$$

where the total scalar irradiance $E_0 = E_{0\uparrow} + E_{0\downarrow}$. From (1) and (2), we use the model output to calculate local spectral heating 30  rates as

$$H_\lambda = a(|E_{0\uparrow}| + |E_{0\downarrow}|). \tag{3}$$

Bulk transmittance over the range of wavelengths from $\lambda_1$ to $\lambda_2$ is defined as

$$T_{\text{bulk}} = \frac{\int_{\lambda_1}^{\lambda_2} T(\lambda) E_{\text{in}}(\lambda) \, \mathrm{d}\lambda}{\int_{\lambda_1}^{\lambda_2} E_{\text{in}}(\lambda) \, \mathrm{d}\lambda} = \frac{\int_{\lambda_1}^{\lambda_2} E_{\text{tra}}(\lambda) \, \mathrm{d}\lambda}{\int_{\lambda_1}^{\lambda_2} E_{\text{in}}(\lambda) \, \mathrm{d}\lambda} \tag{4}$$

where $T$ is spectral transmittance, $E_{\text{in}}$ is incident total irradiance (downward diffuse plus direct beam), and $E_{\text{tra}}$ is transmitted total irradiance. Unless otherwise specified, the wavelength range over which the bulk transmittance is calculated is 400–900 nm. Bulk albedo is calculated similarly, but with reflected irradiance instead of transmitted irradiance in the numerator.

## 2.4   Model setup

Irradiance was calculated every 2 nm, and the resulting spectra smoothed with a Gaussian filter, to approximate the spectral resolution of the RAMSES sensors.

Model parameters were chosen through an iterative process to obtain a good correspondence with the measured data.

In the radiative transfer model, the sea ice under ponds is represented by two layers (fig. 2). The upper layer has a thickness of 10 cm, with an air volume fraction that depends on the average pixel intensity of the aerial photo (see below), with higher air volume fractions for brighter areas. Brighter areas appeared to be caused by a layer containing a larger amount of air, but the thickness of this layer in the simulations (10 cm) was chosen arbitrarily. The lower layer extends from 10 cm below the pond to the measured ice thickness, and has a fixed air volume fraction of 0.1%. Bare ice is also represented with two layers, but here the thickness of the upper layer, corresponding to the surface scattering layer (SSL), is determined by the measured freeboard. The upper layer, which consisted of granular ice, was simulated as large-grained snow, i.e. ice spheres in air, with a radius of 2.5 mm. All ice, with the exception of the SSL, was simulated with a 20% brine volume fraction. The effective radius of air bubbles in all simulations was set to 0.25 mm, which is a bit higher than the upper bound for the air inclusions found by Light et al. (2003), though it should be noted that was for ice at a temperature of $-15°$C, whereas the ice in our study was warmer, around $-1°$C. For brine, the effective radius of inclusions (1.5 mm) was taken to be constant for all ice layers in the simulations.

Values of the effective scattering coefficient, $b_{\text{eff}} = b(1 - g)$, are shown in fig. 2. Light et al. (2015) used $b_{\text{eff}}$ of $102 \, \text{m}^{-1}$, $15 \, \text{m}^{-1}$ and $\approx 1 \, \text{m}^{-1}$ in the SSL, drained layer and interior ice, respectively. Hence, our SSL has less scattering than that of Light et al. (2015), but we did not include a drained layer. On the other hand, our interior ice has higher scattering. For a review of $b_{\text{eff}}$ used in various publications, see table 2 in Petrich et al. (2012).

Ponds are represented by adding a layer of pure sea water on top of the sea ice, with the thickness of the water layer equaling the measured pond depth. Water depth in the region where the measurements were taken is around 3 km, and that depth is used as the total water depth in the radiative transfer simulations. The sea water itself was pure sea water with the addition of absorption and scattering measured during the Norwegian Young Sea Ice Cruise in 2015 ("pre-bloom" case from Taskjelle et al., 2017a). The solar zenith angle was set corresponding to the time and approximate location of measurements, with values between $64.5°$ and $66.3°$.

The number of streams used in the upper slab, i.e. the atmosphere, was set to $N_{\text{U}} = 32$. The number of streams in the lower slab (ice and ocean) is set automatically by AccuRT to $N_{\text{L}} = N_{\text{U}} \cdot m_{\text{max}}^2$, where $m_{\text{max}}$ is the maximum refractive index of the

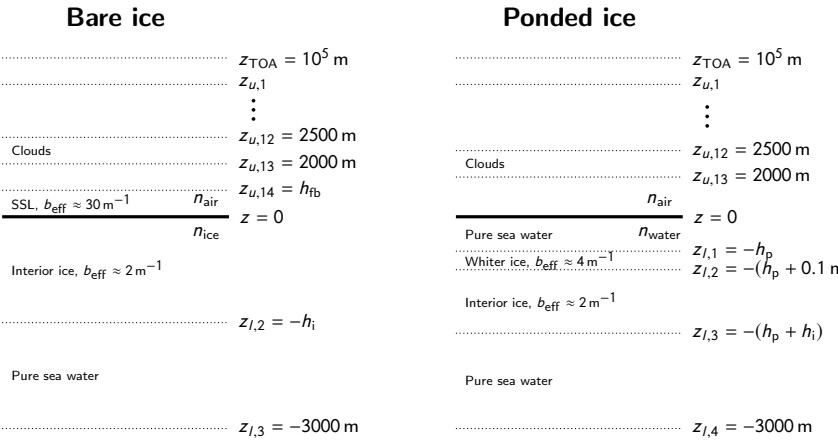

**Figure 2.** Sketch of layers in AccuRT. Values for the effective scattering coefficient $b_{eff} = b(1-g)$ are averages over the visible range.

**Table 1.** Measurement summary for all measurements along the transects. $\bar{h}_p$ is mean pond depth, $\bar{h}_i$ is mean ice thickness, $\bar{h}_{fb}$ is mean freeboard, and $\bar{T}$ is mean bulk transmittance [eq. (4)] in the range $400\,nm$ to $900\,nm$. Numbers in parenthesis indicate one standard deviation. $N_p/N_b$ is the number of spectra collected along the transect under ponded and bare ice, respectively. Positions are approximate.

| Date | Position | $N_p/N_b$ | Ponded ice | | | Bare ice | | |
| | | | $\bar{h}_p$ [m] | $\bar{h}_i$ [m] | $\bar{T}$ | $\bar{h}_{fb}$ [m] | $\bar{h}_i$ [m] | $\bar{T}$ |
| --- | --- | --- | --- | --- | --- | --- | --- | --- |
| T1: 27.07.2012 | 82.4°N 20.8°E | 15/15 | 0.12(0.05) | 0.59(0.05) | 0.39(0.10) | 0.12(0.05) | 0.88(0.15) | 0.15(0.09) |
| T2: 30.07.2012 | 82.35°N 21.5°E | 17/17 | 0.21(0.07) | 0.54(0.10) | 0.46(0.10) | 0.11(0.04) | 0.86(0.09) | 0.20(0.10) |
| T3: 01.08.2012 | 82.1°N 21.9°E | 16/19 | 0.15(0.10) | 0.47(0.07) | 0.39(0.10) | 0.14(0.05) | 0.71(0.10) | 0.17(0.12) |

lower slab in the simulated wavelength range, which gives $N_L = 57$ for ponded ice and 61 for bare ice. These high numbers of streams were needed to avoid a numerical artifact in the calculation of albedo over the highly scattering bare ice.

Varying cloud cover was present on all days, causing large variations in incident irradiance. Integrated from 400nm to 900nm, values ranged from $84\,W\,m^{-2}$ to $266\,W\,m^{-2}$. In all simulations clouds are represented by a $0.5\,km$ thick layer of water droplets, with a base height of $2\,km$, and an effective droplet radius of $10\,\mu m$. The volume fraction of cloud particles was adjusted so that the simulated incident irradiance at $460\,nm$ was within about $5\,mW\,m^{-2}\,nm^{-1}$ of the measured incident irradiance. The applied volume fractions ranged between $1.3 \times 10^{-10}$ and $9.1 \times 10^{-7}$, corresponding to optical depths between 0.007 and 45.7.

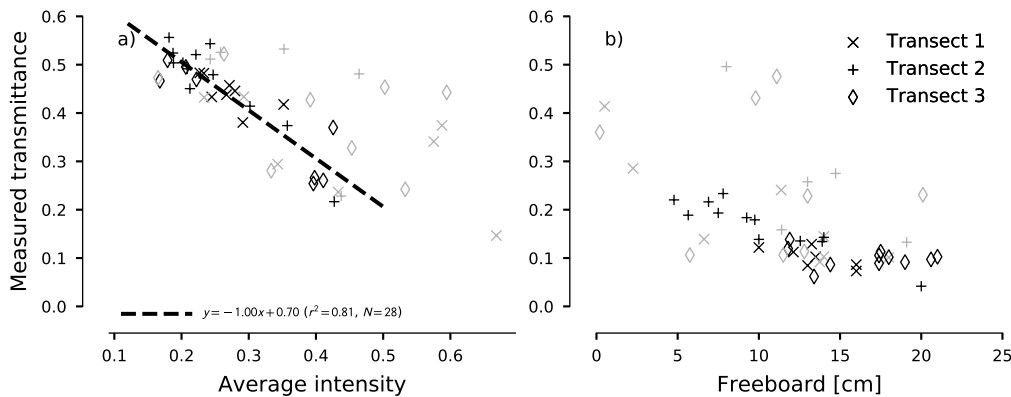

**Figure 3.** a) Measured bulk transmittance under ponds versus surface average intensity determined from an aerial photo (cf. fig. 4). The dashed line is a regression lines for only locations away from pond edges (black markers). b) Measured bulk transmittance versus measured freeboard. For both panels, the light grey markers are for locations close to the boundary between ponds and bare ice.

### 2.4.1 Parameterizing air content

To investigate whether the surface properties as seen from above can give us any useful information, we look at an average intensity of the RGB aerial photo (fig. 1), calculated as the mean of the three channels (red, green and blue) divided by 255 to obtain a value between 0 and 1, similar to Katlein et al. (2015a). The along-transect average intensity, shown in fig. 4, is calculated as follows. For each pixel along the approximated center line of the transect, a weighted average of a 21 pixel wide square around the pixel was calculated. The weights were given by a Gaussian function, with a standard deviation corresponding to 5 pixels, centered on the middle pixel.

Average intensity is seen to have a clear negative correlation with transmittance for the measurements in ponds away from pond edges (fig. 3a), an observation that is used to obtain an air volume fraction for the upper $10\,\mathrm{cm}$ ice as follows. A series of simulations were performed with different pond depths, ice thicknesses and air bubble volume fractions for the upper $10\,\mathrm{cm}$ of ice, for the wavelength range $400\,\mathrm{nm}$ to $900\,\mathrm{nm}$. For a given pond depth and ice thickness, bulk transmittance (eq. (4)) was seen to decrease exponentially with increasing air volume fraction, and an exponential curve of the form $a \cdot e^{-b\phi} + c$ was fitted to the points using the `curve_fit` function from `scipy.optimize`, giving transmittance as a function of air volume fraction $\phi$. The coefficients $a$, $b$, and $c$ could each be well described using a quadratic dependence on ice thickness, while pond depth was found not to influence the values of the coefficients significantly. Further, the measured ice thickness was used to calculate the coefficients for the exponential curve. Finally, an air volume fraction for the upper $10\,\mathrm{cm}$ of ice was calculated by equating the linear fit from fig. 3a with the exponential curve, inserting the average intensity and solving for $\phi$.

For the locations close to pond edges (grey symbols in fig. 3a), the median air volume fraction was used.

For the bare ice (fig. 3b) we can see that a larger freeboard generally leads to lower transmittance, for freeboards up to about $15\,\mathrm{cm}$. For freeboard larger than $15\,\mathrm{cm}$ there is no clear trend in transmittance. Beyond using the measured freeboard as the thickness of the high-scattering layer, no adjustments are made similar to those made for the ponded ice.

## 3   Results

Figure 4 shows measured ice thickness, pond depth and bulk transmittance, along all three transects, as well as extracts from an aerial photo showing the surface in a narrow region along the transects (cf. fig. 3). The bulk transmittance is calculated for the spectral range 400–900 nm, with values summarized in table 1. As expected, transmittance is generally higher through the darker ponds, where the ice thickness is generally smaller than outside ponds, and the surface has a lower albedo.

The extracts from an aerial photo displayed in fig. 4 are from an image captured on 28 July, i.e. between the days of the first two transects (Table 1). These extracts represent a 10 pixel wide strip, corresponding to about 1.6 m, or nearly $2h_i$, along the line of the transect.

Simulated broadband albedo from 350 to 2200 nm for the bare ice cases is distributed around a mode of 0.62, which is a bit higher than the white ice albedo of 0.55 reported by Hudson et al. (2013) for the same wavelength range.

Table 1 shows mean values of pond depth, ice thickness and measured bulk transmittance (400–900 nm) along each transect, as well as date and approximate position, and number of irradiance measurements below the ice.

Parts of the ponds have a distinctly brighter appearance (fig. 1), possibly caused by a layer containing a larger amount of air bubbles. Figure 3 shows the bulk transmittance of ponded ice versus the average intensity of the surface, obtained from the aerial photo (figs. 1 and 4). Considering only the points that are away from pond edges (black markers), there are two clusters of points. One cluster is centered on a average intensity of about 0.2 and transmittance of 0.5, the other, smaller cluster is at a average intensity of 0.4 and transmittance of 0.25. These two clusters represent 'dark' and 'light' ponds, respectively. The points found in the second cluster are the first point (0 m) on transect 2, and the first three (0–2.2 m) on transect 3, cf. fig. 4.

A comparison of measured and simulated values is seen in fig. 5, which shows bulk transmittance [eq. (4)] for all locations in the three transects. We see that the outliers generally represent the locations that are identified as being close to a pond edge (cf. §2.1), marked by grey circles in figs. 4 and 5.

The distance to the pond edge is the along-transect distance, calculated from the measured pond depth and freeboard. In addition to the locations found by that criterion, point number 11 (11.25 m) of transect 1, and points number 12 (12.1 m) and 28 (29.7 m) along transect 3 are also identified as being close to a pond. For the point on transect 1, this classification is due to it being far away from the regression line in fig. 3a. For point 12 on transect 3, this classification is due to the notably higher transmittance seen here compared to the surrounding locations, likely caused by the pond adjacent to the transect, that can be seen in fig. 1. Similarly, point 28, which is a ponded location, seems to be influenced by bare ice close to it.

The distribution of simulated transmittance is bimodal, with one mode corresponding to ponded ice, the other mainly to bare ice. Measured transmittance has a trimodal distribution, where the third mode may be at least partly a result of the lighter blue ponds. For the simulated transmittance, edge cases will have been shifted towards the one of the modes, while the data points corresponding to light ponds are part of the mode related to bare ice, due to their lower transmittance.

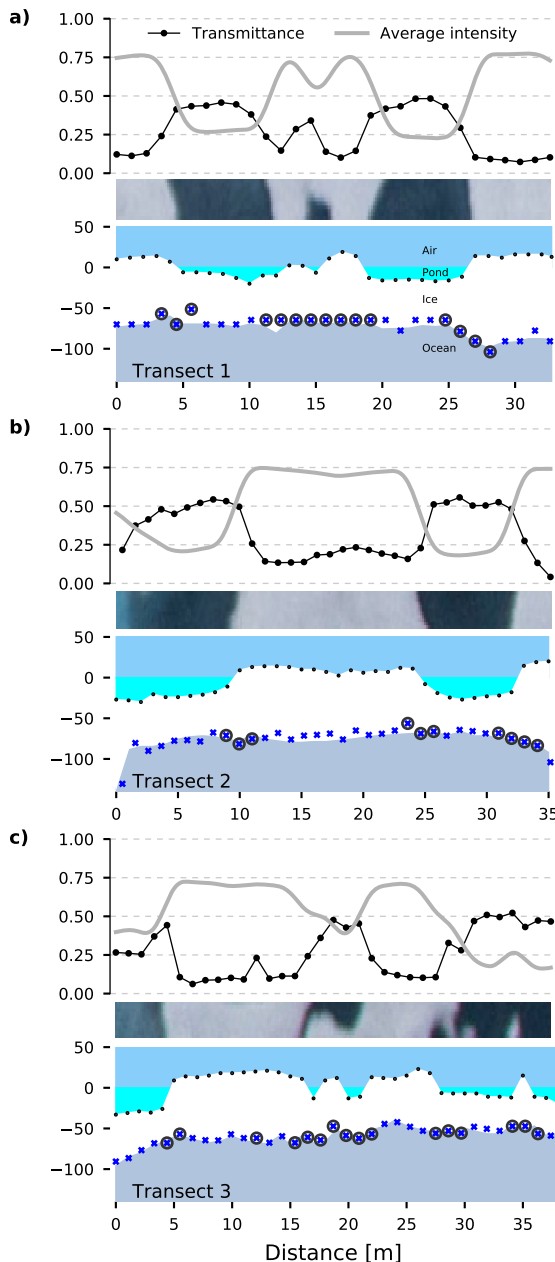

**Figure 4.** Bulk transmittance and average intensity of ice (upper subpanels), aerial photo (middle subpanels), and pond depth and ice thickness (lower subpanels), for a) transect 1, b) transect 2, and c) transect 3. The average intensity is calculated as described in Sect. 2.4.1. The profile of ice thickness and pond depth is based on measurements of ice thickness, pond depth and freeboard performed along the three transects. Black dots along the ice surface in the lower panels indicate locations of thickness drillings, blue crosses near the ice bottom indicate the locations of under-ice irradiance measurements, with the depth as given by the pressure sensor of the RAMSES instrument. Circled blue crosses indicate measurements that are identified as being close to pond edges.

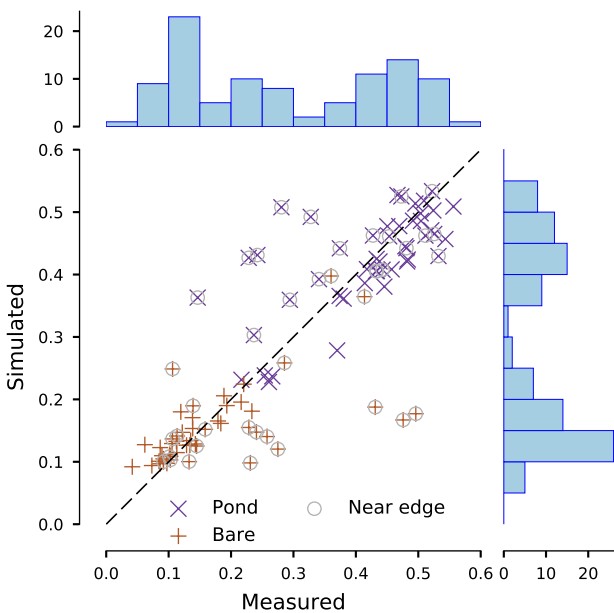

**Figure 5.** Simulated values versus measured values of bulk transmittance [eq. (4)] (400 nm to 900 nm) for all three transects. The black dashed line indicates a 1:1 correspondance. Blue crosses are for ponded ice, red plus signs are for bare ice. Data from locations close to edges of ponds are marked with grey circles. On top is a histogram for observed values, on the right for simulated values.

## 4 Discussion

### 4.1 Observations

Some other transmittance data from the same year and region can be found in the literature. The bare ice transmittance in our study is comparable to the measurements by Wang et al. (2014) (transmittance of ∼0.2 in mid-July), though this was for a different wavelength region (350–800 nm). Katlein et al. (2015b) measured transmittance up to about 0.3 in mid-August, but used a wider wavelength region (320–950 nm).

Not limited by location and time, there are multiple other studies to be found. Katlein et al. (2015a), also for the range 320–950 nm, showed transmittances that were generally below 0.1, but this was for thicker ice. Our PAR (photosynthetically active radiation, 400–700 nm) transmittances are similar to those measured by Ehn et al. (2011), but generally somewhat higher for bare ice, likely due thinner ice in our study. Where we have PAR transmittance of 0.20–0.25 for bare ice and 0.50–0.59 for ponded ice, Ehn et al. (2011) reported 0.07–0.20 and 0.34–0.65 for bare and ponded ice, respectively. Our results are also in upper range of those reported by Light et al. (2015), where PAR transmittance was 0.03–0.22 (bare ice) and 0.13–0.58 (ponded ice). Nicolaus and Katlein (2013) showed PAR transmittances no higher than 0.4, for a study comprising several ROV deployments at different locations in the Arctic ocean. That the transmittance in this study is high compared to other studies is likely due to this being relatively thin first year ice.

Concerning the ice thickness and pond depth used in simulations, there are several factors that give rise to uncertainties regarding these. First, the rope was located about $1\,\mathrm{m}$ below the ice, but the diver held the radiometer at the ice bottom. The diver therefore had to estimate the position at the ice bottom to be as close as possible to the mark on the rope. Second, rope stretching makes the distances of the mark on the rope from the start of the transect an estimate. Finally, the rope stretching also means that the under-ice irradiance measurements were not performed in the same locations as the thickness drillings, with differences possibly up to $0.5\,\mathrm{m}$. To account for the rope stretching, ice thickness, pond depth and freeboard were interpolated linearly to the estimated locations of the irradiance measurements (fig. 4), though there remains some uncertainty in the exact locations of both the irradiance measurements and the thickness measurements.

The aerial photo (fig. 1) from which the along-transect average intensity has been determined was captured on 28 July, i.e. one day after transect 1 was sampled, 2 days before transect 2 was sampled, and 3 days before transect 3 was sampled. As the surface may have changed somewhat in the interim, the average intensity obtained from the image may not exactly represent the conditions at the time of the radiometric measurements.

## 4.2 Spectral shape of transmittance

As a different way of looking at how the model performs, we take a closer look at the spectral shape of measured and simulated transmittance. Figure 6 shows the mean gradient of the transmittance spectra ($\mathrm{d}T/\mathrm{d}\lambda$), for bare and ponded ice separately, calculated using `numpy`'s `gradient` method. While there is generally a good correspondence between gradients from observed and simulated transmittance spectra, some differences stand out. At around $730\,\mathrm{nm}$ the observed spectra have a notably steeper gradient for ponded ice, whereas the bare ice fits better. $730\,\mathrm{nm}$ is a wavelength where the absorption coefficient of pure water is higher than that of pure ice, and a higher brine volume gives a steeper gradient here. As such, this difference between simulated and observed data could indicate that the specified brine volume of 20% is too low.

The troughs in the gradient seen near $690\,\mathrm{nm}$ and $760\,\mathrm{nm}$ are likely caused by absorption by atmospheric oxygen, which changes the angular distribution of the light field, and therefore may give a local increase of transmittance as described in Taskjelle et al. (2016).

Near $800\,\mathrm{nm}$ the gradient for observed spectra exceeds zero, whereas the gradient for simulated spectra is always negative. This discrepancy, which is particularly visible in the case of ponded ice, was discussed in Taskjelle et al. (2016), but no satisfying explanation was found.

## 4.3 Model setup

The simulated ice has an air volume fraction (0.0015) that is about an order of magnitude smaller than that of the ice studied by Hamre et al. (2004). At the same time, the effective radius of $0.2\,\mathrm{mm}$ used here is much smaller than the $0.93\,\mathrm{mm}$ used by Hamre et al. (2004) for fast ice in Kongsfjorden, Svalbard, but more consistent with the first-year fast ice from Point Barrow, Alaska, studied by Light et al. (2003). Compared to Hamre et al. (2004), the differing volume fraction and effective radius has opposite effects on transmittance. The smaller air volume fraction leads to less scattering, but a smaller radius gives a higher number of air bubbles, which will increase scattering, with the scattering efficiency peaking for sizes comparable to the

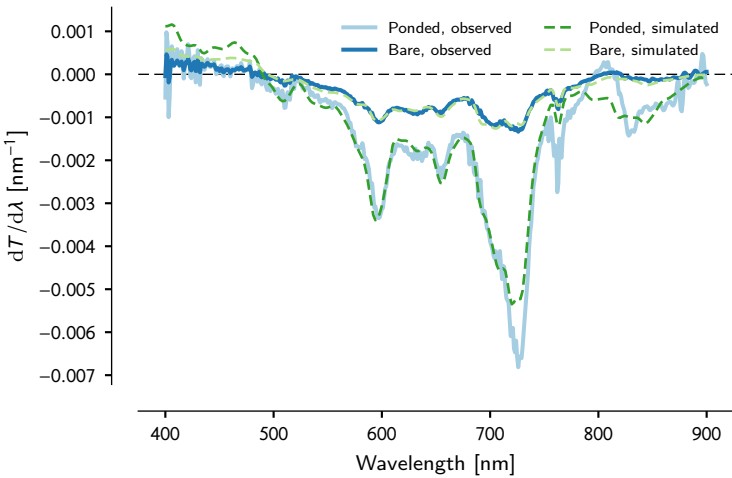

**Figure 6.** Mean gradients ($dT/d\lambda$) of simulated and observed spectra, for bare and ponded ice. Only locations that are identified as being away from pond edges are included.

wavelength of the incident light. That smaller spheres with constant volume fraction increases scattering is true to a certain point, for very small spheres scattering approaches the Rayleigh regime with a very small cross section per particle.

It should be noted that brine inclusions in sea ice are generally not spherical (Light et al., 2003). Light et al. (2003) related an equivalent spherical radius to the length of brine inclusions, but only for ice at a temperature of $-15°$C. To represent both
absorption and scattering accurately, both total volume and total area need to be the same for the spheres as for the original inclusions. Light et al. (2003) shows (their fig. 9a) that for a brine inclusion length of $3\,\text{mm}$, the radius of equivalent spheres, conserving both area and volume, is about $0.1\,\text{mm}$, while for a brine inclusion length of $1\,\text{cm}$, the equivalent sphere radius increases to about $0.15\,\text{mm}$. The ice in our study was much warmer than $-15°$, so brine inclusions are likely comparatively large. Hence, even when using the highest equivalent radius from Light et al. (2003), $0.15\,\text{mm}$, for brine inclusions in AccuRT,
while keeping the brine volume at 20%, the scattering increases so much that, even with zero air bubbles, transmittance decreases. However, this likely just indicates that the properties of our ice are very different from the colder FYI studied by Light et al. (2003), so that a larger equivalent radius is more appropriate.

The ice was in a late stage of melt, with temperatures near $0\,°$ at the top and around $-1.2\,°$ at the bottom. On the bare ice regions there was a surface scattering layer above the freeboard, which consisted of deteriorated ice that was in a granular
form, similar to coarse grained snow. Figure 7 shows that the interior ice was visibly porous, with some deteriorated internal layers. The ponds on the surface had drained to sea level, indicating that most brine pockets or channels would have been filled with melt and/or sea water. Detailed structural analysis and identification of air- versus liquid-filled inclusions was hampered by the immediate drainage of the liquid inclusions when a core was taken up, but it is safe to say that the ice did not retain the traditional structure of sea ice, determined by freezing equilibrium relationships. As a result of the heavily modified state of
the ice, our estimates of brine and air volumes and sizes may be highly uncertain.

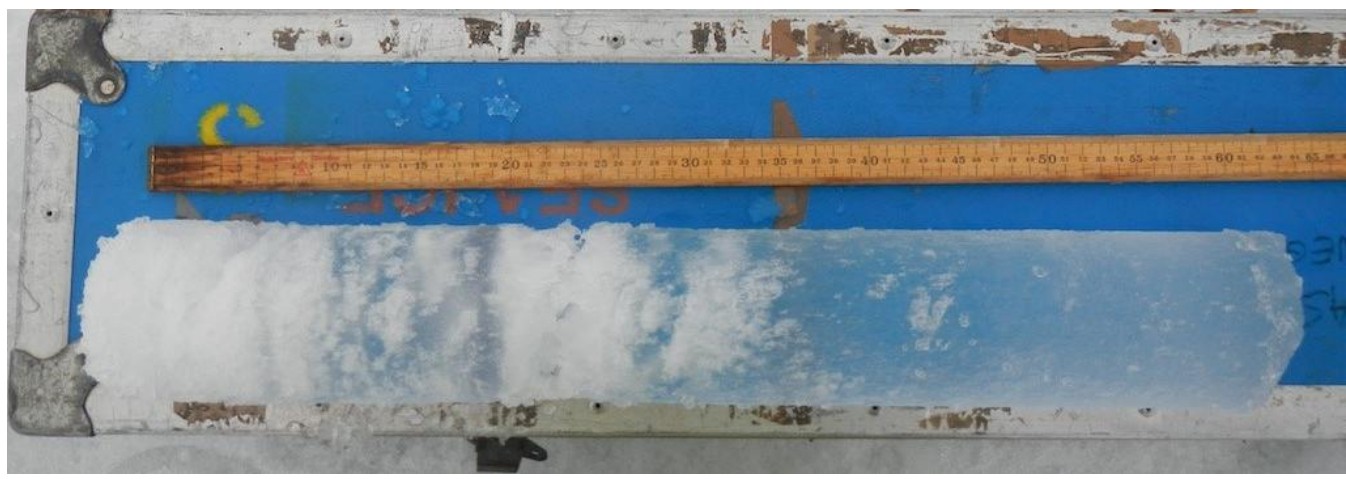

**Figure 7.** Photo of an ice core extracted from bare ice near the dive hole of the third transect.

Further, the vertical orientation of brine inclusions commonly seen in sea ice, appears to cause anisotropic scattering in the ice (Trodahl et al., 1989; Katlein et al., 2014), which AccuRT cannot account for. This issue may be important for determining the exact radius over which edge effects are important.

As little information about the vertical structure of the ice is available, the number of layers in the model is kept at a
minimum. While assumptions could be made based on typical profiles of salinity, for the sake of simplicity, and reducing the number of assumptions, we have chosen to not do this.

### 4.4 Average transmittance

The results in fig. 5 are what one might expect – simulations of bare ice underestimates the transmittance when compared to measurements near a pond, and simulations of ponded ice overestimate the transmittance when near bare ice. However, due to
these opposing effects, the averaged bulk transmittances are similar, being 0.28 and 0.29 for simulated and measured values, respectively, including outliers. Hence, for area-wide averages the edge effects may largely cancel each other out.

Looking at a simpler case, we simulate a typical ponded and bare ice case for each transect, using the mean pond depth, freeboard and ice thickness. Ice properties are as described in Sect. 2.4, with the amount of air in the upper $10\,\mathrm{cm}$ of the under-pond ice as the median of the values found based on the luminosity (Sect. 2.4.1). The total length of each surface type is
found (cf. fig. 4), and a length-weighted average transmittance is calculated using

$$\bar{T} = \frac{L_\mathrm{p}T_\mathrm{p} + L_\mathrm{b}T_\mathrm{b}}{L_\mathrm{p} + L_\mathrm{b}}, \tag{5}$$

where $L_\mathrm{p}$ and $L_\mathrm{b}$ is the length of the transect covered by ponds or bare ice, respectively, and $T_\mathrm{p}$, $T_\mathrm{b}$ is the corresponding transmittances. This approach can easily be applied to area fractions of surface types by replacing the lengths by areas.

The mean measured bulk transmittances for the three transects are respectively, 0.27, 0.33 and 0.27 for transects 1, 2 and
3. Using (5) and transmittance based on simulations using mean values, we get 0.30, 0.30 and 0.29 for the three transects

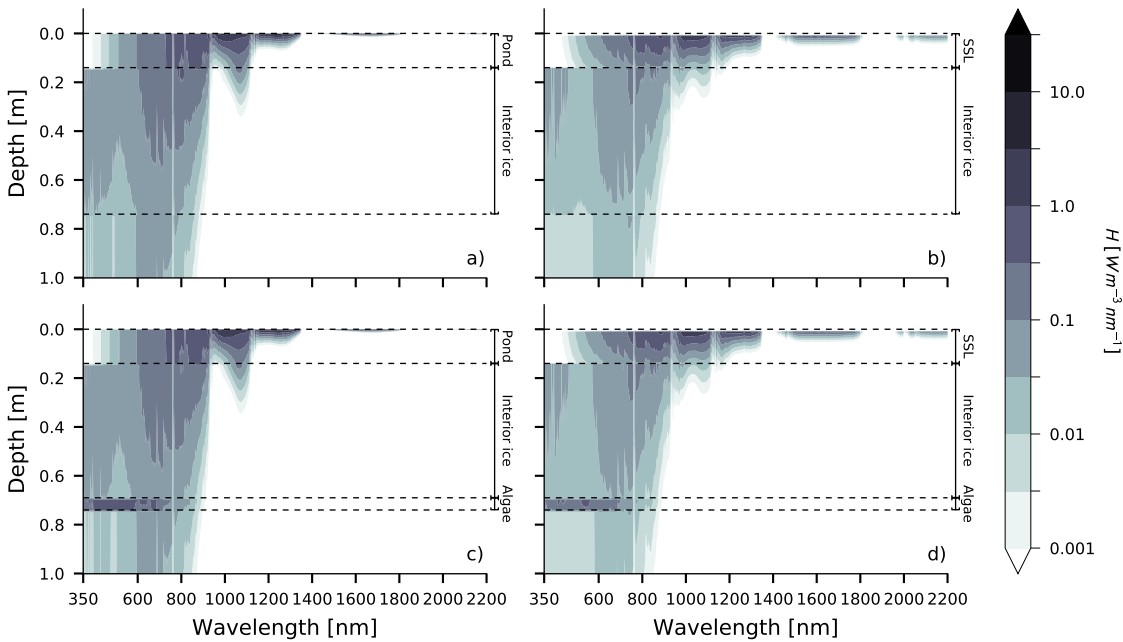

**Figure 8.** Absorbed energy calculated from simulated spectral irradiance using Gershun's law, for ponded ice (a, c) and bare ice (b, d). In c) and d) a Chlorophyll concentration of $500\,\mathrm{mg\,m^{-3}}$ is added to the bottom $5\,\mathrm{cm}$ of the ice.

individually, and $0.29$ for all points. While the average simulated values correspond well to measured values, one should note that the ponds in this study are generally wide compared to the ice thickness. A similar correspondence would not necessarily have been seen if the ponds were so small, or the ice so thick, that the light field beneath most or all of the ponded area were affected by the surrounding bare ice.

## 4.5 Energy absorption

Figure 8 shows contour plots of the spectral heating rate $H_\lambda$, which is the energy absorbed per unit volume per wavelength per unit time, given by (3). The absorbed energy is calculated from the simulated scalar irradiance field and total absorption coefficient of the layer. The scalar irradiance field was calculated every $1\,\mathrm{cm}$ in the pond and ice. For easier comparisons, layer thicknesses were equal for both ponded and bare ice. I.e., the thickness of the SSL was set equal to the pond depth ($14\,\mathrm{cm}$), and the thickness of the interior ice was $60\,\mathrm{cm}$ in both cases, giving a total SSL/pond + interior ice thickness of $74\,\mathrm{cm}$.

Due to the higher absorption coefficient of both ice and water in the near-infrared range compared to the visible range (Segelstein, 1981; Warren and Brandt, 2008), the near-infrared has the highest surface heating rate. Beyond $1400\,\mathrm{nm}$ nearly all the energy is absorbed in a layer just a few centimetres thick. In the visible range, where the incident irradiance is higher, the absorption is significantly lower, meaning that more energy is deposited deeper in the pond-ice-system, and in the ocean.

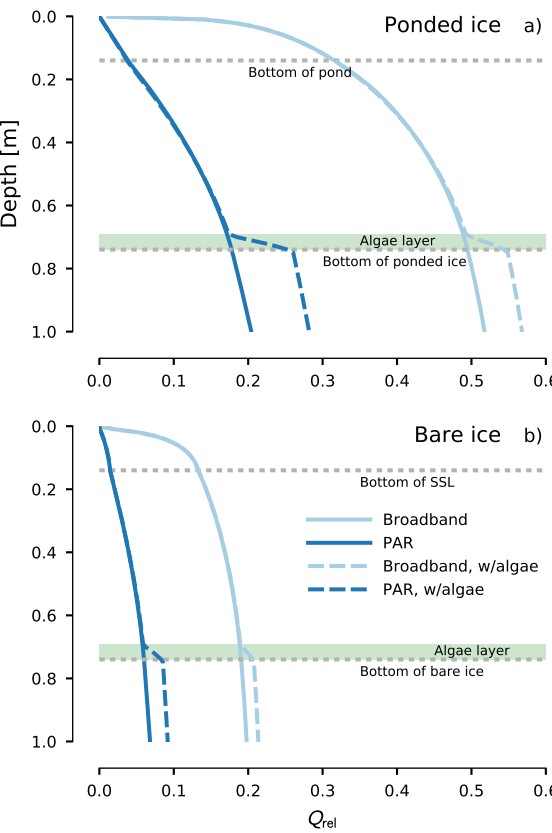

**Figure 9.** Cumulative heating (eq. (6)) relative to the incident irradiance, for a) ponded ice (fig. 8a,c), and b) bare ice (fig. 8b,d). The green shaded region indicates where Chlorophyll $a$ is added for the algea cases. Solid lines are without algae in the lower $5\,\mathrm{cm}$, dashed lines are with algae. Darker lines for the PAR range, lighter colored lines for broadband (350–2200 nm).

High concentrations of algae in the bottom layer of the ice could influence the energy deposition (Ehn and Mundy, 2013). In fig. 8c,d we show the result of a simulation where absorption and scattering corresponding to $500\,\mathrm{mg\,m^{-3}}$ Chlorophyll $a$, as described in Hamre et al. (2004), is added to the bottom $5\,\mathrm{cm}$ of the ice. Welch and Bergmann (1989) measured Chlorophyll $a$ concentrations of over $5000\,\mathrm{mg\,m^{-3}}$ in the Canadian Arctic, albeit for $2.5\,\mathrm{cm}$ ice, thus $500\,\mathrm{mg\,m^{-3}}$ is a realistic concentration

5    in sea ice. Figure 9 shows the corresponding profiles of cumulative absorption. Note that the diffuse attenuation coefficient for PAR in the algae layer is around $5\,\mathrm{m^{-1}}$, which is significantly lower than the $\sim22\mathrm{m^{-1}}$ estimated by Ehn and Mundy (2013) for the same Chlorophyll $a$ concentration.

To compare the amount of heating within the ice for the two cases, fig. 9 shows the cumulative heating of the ice (or ice with pond) relative to the incident irradiance, $Q_{\mathrm{rel}}$, for both the range 350–2200 nm, and the PAR range. The heating rate eq. (3)

10    is first integrated over the wavelength range, and then cumulatively integrated from the surface and downwards. This result is divided by the energy entering the ice $E_{\downarrow}$, i.e. the downward planar irradiance less the specularly reflected irradiance just above

the atmosphere-ice–interface. Hence,

$$Q_{\text{rel}}(z) = \frac{\int_0^z H(z')\,\mathrm{d}z'}{\int_{\lambda_1}^{\lambda_2} E_\downarrow \,\mathrm{d}\lambda}, \tag{6}$$

where $H(z') = \int_{\lambda_1}^{\lambda_2} H_\lambda(\lambda, z')\,\mathrm{d}\lambda$.

We see in fig. 9 that about $19\%$ of the incident energy is absorbed in the bare ice, and $49\%$ in the pond and ice below it. Considering all 99 cases corresponding to measurements, the mean (standard deviation) is $22\%$ ($6\%$) and $51\%$ ($3\%$) for bare and ponded ice, respectively. Considering only the SSL and pond, the values are $15\%$ ($5\%$) and $35\%$ ($4\%$). With algae present the values are $21\%$ and $54\%$, respectively. That a smaller fraction of the incident energy is absorbed in the bare ice is caused by the higher amount of light scattered back to the atmosphere from the surface scattering layer. In general, a thicker SSL in the simulations give a higher albedo, and a smaller fraction of energy absorbed in the ice. Simulations from ponded ice show no clear correlations between the various parameters, such as pond depth, ice thickness, albedo and absorbed energy.

On another note, some of this backscattered light is again scattered back towards the surface when it hits the thin cloud layer present in both simulations. As a result of this interaction with the clouds, the simulated incident irradiance is $45\%$ higher over the bare ice compared to the darker ponded surface ($213\,\mathrm{W\,m^{-2}}$ vs. $147\,\mathrm{W\,m^{-2}}$). Hence, where the modelled relative absorption is equal for bare and ponded ice, the bare ice has a higher absolute absorption. This effect of higher incident irradiance over bare ice will likely not be present at all in reality, as the horizontal scale of pond features is small, on the order of a few meters. A radiation enhancement effect due to scattering between surface and clouds may be present, but the area scattering radiation back to the clouds will be a mixture of bare ice and ponded ice, so there will likely be very little or no spatial variability of incident irradiance along the transects.

In both cases the strongest heating rate is seen right below the surface, due to the strong absorption by both water and ice in the near-infrared range (cf. figure 8). The absorption and scattering coefficients in the interior ice are the same in both cases, as the properties of the ice are the same. Therefore, the stronger heating rate seen in the interior ice under the pond is caused by the higher amount of available light.

## 5  Conclusions

Transmittance measurements of ponded first-year Arctic sea ice were carried out north of Svalbard, in summer 2012. Under-ice irradiance was measured approximately every meter along three transects covering both ponded ice and bare ice, demonstrating how transmittance may vary near edges of ponds. Significantly higher transmittance was seen through melt ponds and the underlying ice, than through the adjacent bare ice, due to the highly scattering surface layer of the bare ice. Radiative transfer simulations showed that for locations within a few meters from the boundary between bare and ponded ice, plane parallel models do not perform well. Due to strong horizontal gradients in the sea ice surface properties, the modelled transmittance tends to be too low under bare ice near ponds, and too high under ponds near bare ice, which is expected from an one-dimensional radiative transfer model applied to areas close to a pond boundary. However, when using average values for ice properties to simulated typical cases for bare and ponded ice, the resulting length-weighted average bulk transmittance was

close to the average measured bulk transmittance. This lends further support to a conclusion of Ehn et al. (2011), that one can estimate transmittance over larger areas using typical transmittances for bare and ponded ice, which in turn supports large scale studies like that of Arndt and Nicolaus (2014) and Nicolaus et al. (2012). Our study does not, however, address the case in which the typical pond width is less than twice the ice thickness, so that the entire ponded area is affected by edge effects.

Obtaining information about ponded ice from aerial images as described in section 2.4.1 shows potential for this particular type of study, and similar techniques has been applied successfully in other studies (Divine et al., 2015; Katlein et al., 2015a). Using said information in radiative transfer modelling appear useful in this case, though the exact method might depend on the data that are available, as well as the model itself.

Heating rates calculated from model output show stronger radiative heating in ponded ice, due to the higher albedo of the
bare ice. Cumulative heating rates show that about 50% of the incident radiation is absorbed in ponded ice, and 20% in bare ice. Most of the absorption takes place in the upper few centimeters, due to the high absorption coefficients of ice and water in the near infrared.

Multiple surface-cloud reflections make it difficult to compute downward irradiances accurately over a non-uniform surface, using a one-dimensional model. One possible approach that could be implemented in future plane parallell models, is to use a
surface with an area-averaged albedo, and then apply that incident field as a boundary condition for radiative transfer modelling of ponds and sea ice.

## 6    Data availability

The data (Taskjelle et al., 2017b) is available at https://doi.org/10.21334/npolar.2017.b040b396.

*Acknowledgements.* We thank the crew of R/V *Lance* and the other scientists and engineers on board, in particular Jens Ehn, for their
assistance in carrying out the measurements. We are very grateful to Haakon Hop, Rupert Krapp, Peter Leopold, Michael Tessmann, and Jago Wallenschus for carrying out the diving to measure radiation below the ice. We would also like to thank Knut Stamnes for comments that improved the paper. Funding was provided by the Centre for Ice, Climate and Ecosystems (ICE) at the Norwegian Polar Institute and by the Norwegian Research Council, through the FRINAT program (grants 197236/V30 and 221961/F20). This work was also supported by ACCESS, a European Project within the Ocean of Tomorrow call of the European Commission Seventh Framework Programme, grant
25    265863.

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
