# Peer review of "Modelling radiative transfer through ponded first-year Arctic sea ice with a plane parallel model"

_The Cryosphere, 2017_

## Referee Comment (RC1) · Anonymous Referee #1 · 17 Apr 2017

This manuscript describes observations and modeling of the spectral transmittance of solar radiation penetrating through melting first-year ponded Arctic sea ice. The measurements were carried out over three unique transect lines in the proximity of a single ice station during the month of July. The observations give unprecedented detail about the transmittance of light through a ponded ice cover. The measurements were accomplished with the help of a diver, and as a result of this logistical aid, make up a very valuable data set. The corresponding modeling was done with diligence and care and appears to accurately simulate the observations. The text is clear, concise, and very readable. The figures are generally appropriate and relevant, although almost all of them seem to be missing axis and legend labels. In general, I find this manuscript a

useful contribution to the literature. However, I think substantial revisions are necessary prior to publications.

Major points Equation (2) needs to be derived, or referenced, and motivated. What's the motivation for calculating heating rates in this way? Why is this method chosen? Calculation of the derivative with respect to depth of the net planar irradiance is the more conventional way to estimate heating, but this method of using the scalar irradiance is used instead. Why? What is the advantage of using this method? Are there differences between the two approaches? Figure 2. It would be helpful to place some of the stated b values in context. b = 270 m-1 for the SSL seems small, but the value of g is not stated, so it is difficult to tell, also the geometric thickness of the SSL is assigned the entire freeboard depth. Likewise  $b = 25 \text{ m} \cdot 1$  for the ice interior seems large, but difficult to compare without specifying g. Would be interesting to see how these compare with other published estimates (eq., Light et al., 2008 Fig. 11 and Light et al., 2015 Fig. 8). The need to assume a 20% brine volume fraction (p.5, line 11) for this ice probably means this ice was structurally rotten. This summer ice has likely undergone significant retexturing and excessive brine drainage. This casts doubt on the reliability of a traditional structural-optical model (based on freezing equilibrium relationships), as appears to have been used here. I think it important for the authors to comment on this. I understand that ice microstructural analysis was beyond the scope of this work, but I wonder what the ice looked like? Is there a relevant photograph of an ice core that could be included? Also, on p. 11, 1st paragraph, it is interesting that the authors inferred a much smaller air volume fraction than at least one previous study. Not all air content is the same! Some air will be associated with bubbles formed directly in the ice, some associated with brine inclusions and freezing equilibrium, and some air volume may result from the retexturing of ponded snow. Additionally, some air volume may be associated with the above-freeboard portion of the ice after it drains. Given this, it would be useful to see some discussion about the nature of the air expected in this ice. p.14, line 3-10: The authors do correctly explain that there will in reality be little or no spatial variability of incident irradiance along the transects, so I have trouble
understanding why results treating cloud-surface interactions are even presented in a 1-D modeling study. The histograms shown in Fig. 5 seem interesting, but there is no accompanying discussion about how to interpret them. Figure 6 shows the mean gradient of the simulated and observed spectra, although there is little discussion of what motivated this calculation in the text. I understand that there are differences between the observed and modeled text, but there is no discussion about the physical meaning of these differences. The conclusions section seems weak. The aspects of this manuscript that seem most important are: 1. The high spatial resolution, and comprehensive nature of the observed transmittances along three transects. There was precious little discussion about how these observations compare with previous studies of light transmittance through ponded ice, and there was no discussion about the variability of the observed transmittances. 2. The modeling work, and a discussion of what new contributions are being made. This is not the first time radiative transport through bare and ponded sea ice has been simulated using a 1-D radiative transfer model (e.g., see Light et al., 2008, 2015), so some discussion of what new contributions are being made to our understanding of radiative transfer modeling in this domain seems warranted. I also think the computation of luminosity from the aerial imagery of the ponded ice, and its correlation with ice transmittance is a very useful result.

Minor points Figure 3, : missing axis labels, also legend labels need more description ('Transect'g 1, 2, 3). P4, L14: "Mie-code version", how about "a parameterized Mie model"? Table 1. What are the units for various "h" columns? I assume meters, but it is not stated. Fig. 4: missing y-axis labels. Fig. 5: missing x- and y-axis labels. Fig. 6. Missing x and y-axis labels. And legend labels? p. 10 line 21: without legend labels, and no description in the legend, difficult, if not impossible, to tell which curves correspond to the model, or bare v. ponded cases. p. 12 line 21-22: is that Eqn 2? There needs to be some motivation for using this approach. Fig. 7. Axis labels! Color bar labels? Fig. 8. Axis and legend labels! p. 15, line 15 -16: Seems this is a worthwhile point and should be expanded upon. "Obtaining information about ponded ice from aerial images as described above..."... I presume the authors are referring
to the calculation of luminosity from aerial images, but it would be helpful to, here in the conclusion section, fully summarize the calculation that was performed and comment on the utility of itTwhat worked, what didn't, what would be needed to make this technique viable?

TCD

---

## Referee Comment (RC2) · Anonymous Referee #2 · 24 Apr 2017

The paper "Modelling radiative transfer through ponded first-year Arctic sea ice with a plane parallel model" by Torbjørn Taskjelle et al. describes modelling exercises of sea ice light transmittance based on measurements taken in 2012 in the Arctic. The topic is of high interest for large scale estimation of solar energy fluxes in the Arctic climate system. The manuscript provides a worthy contribution to the ongoing evolution of sea-ice optics, however it falls short on its claims.

The manuscript will need some improvements before publication in the Cryosphere:

My main point of criticism is, that the authors claim to solve the puzzle wether spatial mean values can be correctly represented using plane parallel models. This single case study does not provide this proof, especially as the main critical topic in this case

Interactive
comment

is not even discussed in the manuscript: That plane parallel models can be used to estimate mean values of light transmittance, when the individual patches are much larger than the spatial scale of pond-related edge effects is a rather trivial conclusion. The authors fail to address the critical case, when ponding features have a diameter of less than two times the edge influence radius, when the full pond transmittance value is not reached. At an ice thickness of 1m, this is the case for all features smaller than 4m which might be a significant portion of the ponded landscape.

While I support the use of aerial images as a great tool for spatial analysis of optics in ponded sea ice, some aspects are missing: A very similar approach was used by Katlein et al. (2015, JGR) which should be mentioned in the text. An aerial image is rather a measure of surface albedo, so transmittance should be anti-correlated and not correlated, also ice-thickness should play a role here. Secondly, the authors should double check, wether their use of Luminosity (which seems to be incorrectly used as term for Luminance) – a photometric quantity - makes sense in this purely radiometric context. Sea ice light transmittance, as well as RGB CCD sensitivity is independent from the visual impression of a human eye.

Specific comments: p.1 ll. 2-3: These transmittance values seem rather high compared to other values observed in the same area in the same year. They should be discussed in the context of existing literature later in the text.

Figure 1: Almost all figures are lacking proper axes labels. Here it would be extremely helpful to provide a scale bar or ticked axes. Also the profile lines and names should be presented in a higher contrast.

p.2 ll. 9-10: This is not a general property of ROV surveys. In principle ROV surveys can provide exactly the same positioning directly against the ice with sufficient precision. This is a particular property of the cited surveys but not a general property of ROV surveys.

p.2 l.12: Is it really necessary to abbreviate radiative transfer as RT? Readability in-
creases when writing it out.

p.3 l. 19: Where exactly was this surface sensor? On the ship? On a tripod? Which height? What distance to the site?

p.3. ll.21-22: This procedure is described in Nicolaus et al. (2010, CRST); Currently the sentence is very cryptic for someone not knowing the RAMSES sensor.

p.3. ll.23-28: Why do you mentioned what was NOT done? Maybe this can be left out.

p.3 l.30: The word designate sounds weird in this context.

p.4 l.4-5: This infomation is not as central as the following information about AccuRT so it should be moved down or maybe even left out.

p.4 l.15-17: At this point parameter choices seem very arbitrarily. Maybe one should highlight early on in the paper, that the model parameters were selected according to best fit field observations (and then, how were they selected? Inversion?)

p.4. l.21: It would help if you label the consituents of net irradiance to be planar irradiances.

p.4. l.23: the more common order would be $E\_0$-up as $E\_0$ indicates spherical. If you split it with comma, the 0 could refer to a depth level.

p.5 l.2: I think you mean "spectral resolution" instead of "bandwidth"

p.5. l.4-5: The choice of 10cm of SSL seems fairly arbitrary here (See comment above). The reason for parameter choices should in general be made clearer in the following. Explanations are there but poorly structured.

p.5. l.18: I am surprised, that you evaluate your radiative transfer model down to 3000m. Isn't this very unefficient numerically?

p.6. l.14 ff: I think, that luminosity is the wrong term here (it is not even used in your reference), secondly i do not see the reason why you would want to correct for human

perception. I suggest to stay in the radiometric context and either use mean intensity or you need to weigh the three channels by the incoming values. Also this paragraph lacks any reference to Katlein et al. 2015 who already showed a very similar approach. Also, you seem to ignore any effect of ice thickness here.

Figure 3: Again all axes labels and legend entries are missing

Figure 4: What is the actual width of your presented image stripes? This should be mentioned in the text. Also, cross transect mean luminosity is not the approriate quantity to compare to. Mean intensity should be calculated over a circular footprint area (weighted by radius) to resemble the sensor footprint at the ice surface.

Figure 5: Labels are missing. Your histograms seem very jagged. Likely your bin sizes are too small for this small dataset.

p.10. l.1-2: Edge cases will not amount to a 3rd "edge" mode, but rather show up as a continuous plateau between both modes. This rather looks like you do not have a clear bi-modal distribution on your floe, but rather a 3rd mode which might be related to light blue melt ponds. This is likely evident from an "albedo" histogram of figure 2.

p.10. l.4: "Observation uncertainties"

p.10. l.6-10. While you try to correct for positioning errors on cm scale, i doubt, that your data has actually this precision. Drills are already 5cm thick and just a very slight tilt results in several decimeter error. I doubt, that you have a sub dm accurracy which seems to be described in this paragraph.

p.10 ll. 12-14: You describe the difference in sampling times, but you do not discuss it or even state that you don't think it affects your data.

p.10. ll.15-16: These statements feel more like a method description rather than a discussion. I suggest to move the remnants of this paragraph to the end of the discussion and present important discussion first.

p.10. l.17. ff: The reason and purpose of this paragraph does not become evident. It is certainly valuable, but needs more motivation and explanation.

Figure 6: All labels are missing

p.11 l.10: Obviously AccuRT can not account for anisotropic radiative transfer, however anisotropy is crucial to edge-effects at pond edges, so it should be mentioned also along the discussion of spatial scales (above general comment).

p.12 l. 12ff: This does not account for small spatial scales (see above). Furthermore this equation is written for transects only, but it would improve the usability of this paper if you suggest a form for aereal estimates.

Figure 7: All labels are missing

Figure 8: All lables are missing. What is the relation between PAR and BB and at which depth can we use a conversion constant between them.

p.15 l. 6-7: Your manuscript lacks a comparison of measured fluxes to existing datasets.

p.15 l.14: I suppose it is also very much supporting the works of Arndt & Nicolaus (2014,Cryosphere) and Nicolaus (2012, GRL).

p.15 l.15: This approach is not new. It has been demonstrated in Katlein et al. 2015 and also in some works from the campaign at hand (Divine et al.)

p.15 l.26: Pleas provide an accurate pointer (doi) for the associated dataset. A general pointer to a search engine is not enough.

---

## Author Response (AR1)

**1 Replies to reviewer 1**

*This manuscript describes observations and modeling of the spectral transmittance of solar radiation penetrating through melting first-year ponded Arctic sea ice. The measurements were carried out over three unique transect lines in the proximity of a single ice station during the month of July. The observations give unprecedented detail about the transmittance of light through a ponded ice cover. The measurements were accomplished with the help of a diver, and as a result of this logistical aid, make up a very valuable data set. The corresponding modeling was done with diligence and care and appears to accurately simulate the observations. The text is clear, concise, and very readable. The figures are generally appropriate and relevant, although almost all of them seem to be missing axis and legend labels. In general, I find this manuscript a useful contribution to the literature. However, I think substantial revisions are necessary prior to publications.*

Thank you for the positive initial remarks!

**Major points**

A note on the figures first. Naturally, axis labels and legend text was included in our figures, and they were present in the PDF that was submitted to The Cryosphere (TC). As for why they failed to appear in the discussion paper, I cannot say for sure, but presumably it has to with how fonts are embedded in the figure PDFs. TC takes the submitted PDF and inserts it somehow in a new document containing the page header, and in this process the fonts have apparently disappeared. I should perhaps have noticed myself, but at the same time I do wish you had posted a short comment on the discussion page as soon as you noticed this, and a working PDF could have been provided. I will make sure that the problem does not occur in the published version, should the paper make it to that stage.

> *Equation (2) needs to be derived, or referenced, and motivated. Whats the motivation for calculating heating rates in this way? Why is this method chosen? Calculation of the derivative with respect to depth of the net planar irradiance is the more conventional way to estimate heating, but this method of using the scalar irradiance is used instead. Why? What is the advantage of using this method? Are there differences between the two approaches?*

The two methods are essentially equivalent, as equation 1 states. As for why use scalar irradiance, it is a little more convenient when the scalar irradiance and total absorption are available directly from the model. We could have calculated the change in net planar irradiance between detectors and obtained the mean heating rate for the layer, and the result would have been nearly the same. The only difference is that with scalar irradiance you get a value for a discrete depth, but with planar irradiance you get a value for a layer.

We have rewritten this paragraph so that it hopefully is more clear.

*Figure 2. It would be helpful to place some of the stated b values in context.
b = 270 m-1 for the SSL seems small, but the value of g is not stated, so it
is difficult to tell, also the geometric thickness of the SSL is assigned the entire
freeboard depth. Likewise b = 25 m-1 for the ice interior seems large, but difficult
to compare without specifying g. Would be interesting to see how these compare
with other published estimates (eg., Light et al., 2008 Fig. 11 and Light et al.,
2015*

We have modified the figure to report the effective scattering coefficient, $b_{\mathrm{eff}} = b(1-g)$,
instead, and discussed the values in comparison to in particular Light et al. 2015.

*Fig. 8). The need to assume a 20% brine volume fraction (p.5, line 11) for
this ice probably means this ice was structurally rotten. This summer ice has
likely undergone significant retexturing and excessive brine drainage. This casts
doubt on the reliability of a traditional structural-optical model (based on freezing
equilibrium relationships), as appears to have been used here. I think it important
for the authors to comment on this. I understand that ice microstructural analysis
was beyond the scope of this work, but I wonder what the ice looked like? Is there
a relevant photograph of an ice core that could be included?*

*Also, on p. 11, 1st paragraph, it is interesting that the authors inferred a much
smaller air volume fraction than at least one previous study. Not all air content is
the same! Some air will be associated with bubbles formed directly in the ice, some
associated with brine inclusions and freezing equilibrium, and some air volume
may result from the retexturing of ponded snow. Additionally, some air volume
may be associated with the above-freeboard portion of the ice after it drains. Given
this, it would be useful to see some discussion about the nature of the air expected
in this ice.*

We have added the below discussion of this matter to the manuscript. Figure X is
a new figure showing an ice core extracted near the dive hole for the third transect
(added to end of this document)

"*The ice was in a late stage of melt, with temperatures near 0C at the top and
around −1.2C at the bottom. On the bare ice regions there was a surface scattering
layer above the freeboard, which consisted of deteriorated ice that was in a granular
form, similar to coarse grained snow. Figure X shows that the interior ice was visibly
porous, with some deteriorated internal layers. The ponds on the surface had drained
to sea level, indicating that most brine pockets or channels would have been filled with
melt and/or sea water. Detailed structural analysis and identification of air- versus
liquid-filled inclusions was hampered by the immediate drainage of the liquid inclusions
when a core was taken up, but it is safe to say that the ice did not retain the traditional
structure of sea ice, determined by freezing equilibrium relationships. As a result of
the heavily modified state of the ice, our estimates of brine and air volumes and sizes
may be highly uncertain.*"

*p.14, line 3-10: The authors do correctly explain that there will in reality be lit-
tle or no spatial variability of incident irradiance along the transects, so I have
trouble understanding why results treating cloud-surface interactions are even pre-
sented in a 1-D modeling study.*

Merely to point out that fixed atmospheric conditions in simulations with different surfaces give different incident irradiance, which matters if you'd want to calculate absolute values of energy absorption.

> *The histograms shown in Fig. 5 seem interesting, but there is no accompanying discussion about how to interpret them.*

Some additional comments on this has been added to the corresponding paragraph, which now reads as follows:

*"The distribution of simulated transmittance is bimodal, with one mode corresponding to ponded ice and chiefly to bare ice. Measured transmittance has a trimodal distribution, where the third mode may be at least partly a result of the lighter blue ponds. For the simulated transmittance, edge cases will have been shifted towards the one of the modes, while the data points corresponding to light ponds are part of the mode related to bare ice, due to their lower transmittance."*

> *Figure 6 shows the mean gradient of the simulated and observed spectra, although there is little discussion of what motivated this calculation in the text. I understand that there are differences between the observed and modeled text, but there is no discussion about the physical meaning of these differences.*

Section 4.2 has been rewritten to include some more discussion on this.

> *The conclusions section seems weak. The aspects of this manuscript that seem most important are: 1. The high spatial resolution, and comprehensive nature of the observed transmittances along three transects. There was precious little discussion about how these observations compare with previous studies of light transmittance through ponded ice, and there was no discussion about the variability of the observed transmittances.*

Section 4.1 has been expanded with a comparison to various other studies.

In addition a sentence has been added to the conclusions highlighting the high spatial resolution:

*"Under-ice irradiance was measured approximately every meter along three transects covering both ponded ice and bare ice, demonstrating how transmittance may vary near edges of ponds."*

> *2. The modeling work, and a discussion of what new contributions are being made. This is not the first time radiative transport through bare and ponded sea ice has been simulated using a 1-D radiative transfer model (e.g., see Light et al., 2008, 2015), so some discussion of what new contributions are being made to our understanding of radiative transfer modeling in this domain seems warranted. I also think the computation of luminosity from the aerial imagery of the ponded ice, and its correlation with ice transmittance is a very useful result.*

Whereas e.g. Light 2015 also did modelling for ponded ice, that was for a few select cases, presumably in the middle of ponds. As such, edge effects were not considered. Added another sentence in intro:

*"Neither did Light et al. (2008) or Light et al. (2015) consider edge effects, focusing instead on a few select case studies"*

**Minor points**

> *Figure 3, : missing axis labels, also legend labels need more description (Transectg 1, 2, 3).*

Axis labels were discussed above. There was a *"T"* before the numbers in the legend, this has been changed to *"Transect"* for clarification.

> *P4, L14: Mie-code version, how about a parameterized Mie model?*

This has been rephrased as follows:
*"Their inherent optical properties are calculated using a parameterization based on Mie calculations (Stamnes et al., 2011)."*

> *Table 1. What are the units for various h columns? I assume meters, but it is not stated.*

Correct, that is meters. Units are added.

> *p. 12 line 21-22: is that Eqn 2? There needs to be some motivation for using this approach.*

P12, L21–22: Correct, equation 2. A cross reference is added for clarification. See above for motivation.

> *p. 15, line 15 -16: Seems this is a worthwhile point and should be expanded upon. Obtaining information about ponded ice from aerial images as described above.... I presume the authors are referring to the calculation of luminosity from aerial images, but it would be helpful to, here in the conclusion section, fully summarize the calculation that was performed and comment on the utility of it – what worked, what didnt, what would be needed to make this technique viable?*

Indeed, that is referring the calculation of average intensity (previously luminosity, see comments by the second reviewer). As pointed out by the second reviewer, similar methods has been used in other studies, and these have been mentioned in this context. A full summary of the calculation does not seem that useful here, but we have extended the lines in question as quoted below. An additional caveat is the question of pond size, as remarked by reviewer 2.
*"Obtaining information about ponded ice from aerial images as described in section 2.4.1 shows potential for this particular type of study, and similar techniques has been applied successfully in other studies (Divine et al., 2015; Katlein et al., 2015). Using said information in radiative transfer modelling appear useful in this case, though the exact method might depend on the data that are available, as well as the model itself."*

**2  Response to reviewer 2**

First, regarding the figures. The figures did of course have axis labels and legend entries when we submitted, I believe there were some issues relating to font embedding that caused them to disappear when the header was added by the journal. As I said to the other reviewer as well, I do wish you had posted a short comment on the discussion page, so that a working PDF could have been provided.

**General comments**

> The paper "Modelling radiative transfer through ponded first-year Arctic sea ice with a plane parallel model" by Torbjørn Taskjelle et al. describes modelling exercises of sea ice light transmittance based on measurements taken in 2012 in the Arctic. The topic is of high interest for large scale estimation of solar energy fluxes in the Arctic climate system. The manuscript provides a worthy contribution to the ongoing evolution of sea-ice optics, however it falls short on its claims.

> The manuscript will need some improvements before publication in the Cryosphere: My main point of criticism is, that the authors claim to solve the puzzle wether spatial mean values can be correctly represented using plane parallel models. This single case study does not provide this proof, especially as the main critical topic in this case is not even discussed in the manuscript: That plane parallel models can be used to estimate mean values of light transmittance, when the individual patches are much larger than the spatial scale of pond-related edge effects is a rather trivial conclusion. The authors fail to address the critical case, when ponding features have a diameter of less than two times the edge influence radius, when the full pond transmittance value is not reached. At an ice thickness of 1m, this is the case for all features smaller than 4m which might be a significant portion of the ponded landscape.

It was not our intention to claim a final solution to that puzzle, so perhaps some of the statements were a bit strong. Your final part, regarding ponds that are small relative to the ice thickness, is a very good one. Hence, we have made a couple of additions to the text.

**To the end of section 4.4.:**

"While the average values simulated values correspond well to measured values, one should note that the ponds in this study are generally wide compared to the ice thickness. A similar correspondence would not necessarily have been seen if the ponds were so small, or the ice so thick, that the light field beneath most or all of the ponded area were affected by the surrounding bare ice."

**To conclusion:**

"Our study does not, however, address the case in which the typical pond width is less than twice the ice thickness, so that the entire ponded area is affected by edge effects."

> While I support the use of aerial images as a great tool for spatial analysis of optics in ponded sea ice, some aspects are missing: A very similar approach was

> *used by Katlein et al. (2015, JGR) which should be mentioned in the text. An aerial image is rather a measure of surface albedo, so transmittance should be anti-correlated and not correlated, also ice-thickness should play a role here.*

It is clear from figure 3 that the correlation is negative, but the text is edited to explicitly state so:

"*Average intensity is seen to have a clear negative correlation with transmittance . . .*"

The reference you mention has been added in the first paragraph of section 2.4.1.

> *Secondly, the authors should double check, wether their use of Luminosity (which seems to be incorrectly used as term for Luminance) – a photometric quantity – makes sense in this purely radiometric context. Sea ice light transmittance, as well as RGB CCD sensitivity is independent from the visual impression of a human eye.*

We have changed the method here according to your comments, see the specific comment below.

**Specific comments**

> *p.1 ll. 2-3: These transmittance values seem rather high compared to other values observed in the same area in the same year. They should be discussed in the context of existing literature later in the text.*

Section 4.1 has been extended to include a comparison of several other studies, including Wang et al. (2014) (`http://dx.doi.org/10.1002/2013JC009459`) and Katlein et al (2015) (`http://dx.doi.org/10.1007/s00300-014-1634-3`).

> *Figure 1: Almost all figures are lacking proper axes labels. Here it would be extremely helpful to provide a scale bar or ticked axes. Also the profile lines and names should be presented in a higher contrast.*

Figure labels were discussed above. The profile lines and names have been modified to hopefully make them clearer, and a scale bar added.

> *p.2 ll. 9-10: This is not a general property of ROV surveys. In principle ROV surveys can provide exactly the same positioning directly against the ice with sufficient precision. This is a particular property of the cited surveys but not a general property of ROV surveys.*

Thank you for that insight, the particular sentence has been modified to the following:

"*A disadvantage of some ROV studies is that the ROV is operated too far below the ice on horizontal transects to completely observe local effects.*"

> *p.2 l.12: Is it really necessary to abbreviate radiative transfer as RT? Readability in- creases when writing it out.*

Edited as suggested throughout the manuscript.

*p.3 l. 19: Where exactly was this surface sensor? On the ship? On a tripod? Which height? What distance to the site?*

This paragraph was clumsily written, so has been rewritten. It now starts as:

*"Incident irradiance was measured coincidentally, with the same type of instrument as below the ice, but mounted on a tripod by the dive hole, near the start of the transect. We assume no horizontal variability of incident irradiance between the measuring site, and the transect."*

*p.3. ll.21-22: This procedure is described in Nicolaus et al. (2010, CRST); Currently the sentence is very cryptic for someone not knowing the RAMSES sensor.*

This has been rewritten as:

*"Spectra were collected simultaneously from the two sensors, and as RAMSES sensors are not generally not calibrated to exactly the same wavelengths, the spectra were interpolated to a common wavelength grid, with 1 nm spacing."*

*p.3. ll.23-28: Why do you mentioned what was NOT done? Maybe this can be left out.*

My thought was that readers might ask themselves whether such observations were carried out or not, if no mention of them were made.

*p.3 l.30: The word designate sounds weird in this context.*

I think *"designate"* is appropriate, but we have changed to *"identify"* throughout the manuscript.

*p.4 l.4-5: This infomation is not as central as the following information about AccuRT so it should be moved down or maybe even left out.*

Fair point, this has been moved to the end of the section.

*p.4 l.15-17: At this point parameter choices seem very arbitrarily. Maybe one should highlight early on in the paper, that the model parameters were selected according to best fit field observations (and then, how were they selected? Inversion?)*

I don't understand why this comment refers to these lines, which describe what model output one can obtain, at this point we haven't said anything about parameter choices at all.
But we have added the following at the start of section 2.4:

*"Model parameters were chosen through an iterative process to obtain a good correspondence with the measured data."*

*p.4. l.21: It would help if you label the consituents of net irradiance to be planar irradiances.*

Have specified that it is planar irradiance.

*p.4. l.23: the more common order would be $E_0$-up as $E_0$ indicates spherical. If you split it with comma, the 0 could refer to a depth level.*

Good point, thank you. Edited to $E_{0\uparrow}/E_{0\downarrow}$.

*p.5 l.2: I think you mean "spectral resolution" instead of "bandwidth"*

Edited as suggested.

*p.5. l.4-5: The choice of 10cm of SSL seems fairly arbitrary here (See comment above). The reason for parameter choices should in general be made clearer in the following. Explanations are there but poorly structured.*

This top layer of ice beneath the ponds is not meant to represent a typical SSL composed of granular ice, simply a layer with more bubbles than the deeper ice. Its exact thickness in simulations is however arbitrary, as stated explicitly on line 7 of that page. See also above comment on parameter choices.

*p.5. l.18: I am surprised, that you evaluate your radiative transfer model down to 3000m. Isnt this very unefficient numerically?*

No, with the discrete-ordinates method used here (see e.g. book by Thomas and Stamnes, 1999) the layer depth isn't really significant for computation times.

*p.6. l.14 ff: I think, that luminosity is the wrong term here (it is not even used in your reference), secondly i do not see the reason why you would want to correct for human perception. I suggest to stay in the radiometric context and either use mean intensity or you need to weigh the three channels by the incoming values. Also this paragraph lacks any reference to Katlein et al. 2015 who already showed a very similar approach. Also, you seem to ignore any effect of ice thickness here.*

Using mean intensity is perhaps a better approach, so we have changed the method accordingly. Thank you for the suggestion.

We do not ignore the effect of ice thickness, cf. p. 6 line 22 ff.

*Figure 4: What is the actual width of your presented image stripes? This should be mentioned in the text. Also, cross transect mean luminosity is not the appropriate quantity to compare to. Mean intensity should be calculated over a circular footprint area (weighted by radius) to resemble the sensor footprint at the ice surface.*

It is mentioned in the text (line 13, page 7).

You are of course correct in your second point here, and the method has been modified accordingly. Section 2.4.1 now has the following:

"The along-transect average intensity, shown in fig. 3, is calculated as follows. For each pixel along the approximated center line of the transect, a weighted average of a 21 pixel wide square around the pixel was calculated. The weights were given by a Gaussian function, with a standard deviation corresponding to 5 pixels, centered on the middle pixel."

*Figure 5: Labels are missing. Your histograms seem very jagged. Likely your bin sizes are too small for this small dataset.*

Perhaps you're right. The bin width has been doubled.

*p.10. l.1-2: Edge cases will not amount to a 3rd "edge" mode, but rather show up as a continuous plateau between both modes. This rather looks like you do not have a clear bi-modal distribution on your floe, but rather a 3rd mode which might be related to light blue melt ponds. This is likely evident from an "albedo" histogram of figure 2.*

Thank you for that insight, I can see your point. The paragraph has been rewritten as follows:
" *The distribution of simulated transmittance is bimodal, with one mode corresponding to ponded ice and chiefly to bare ice. Measured transmittance has a trimodal distribution, where the third mode may be at least partly a result of the lighter blue ponds. For the simulated transmittance, edge cases will have been shifted towards the one of the modes, while the data points corresponding to light ponds are part of the mode related to bare ice, due to their lower transmittance.*"

*p.10. l.4: "Observation uncertainties"*

We have in this paragraph added some comparisons with other transmittance measurements, as requested. So while changing the heading as suggested made sense considering the original content of the paragraph, as it is now we'll keep the original heading.

*p.10. l.6-10. While you try to correct for positioning errors on cm scale, i doubt, that your data has actually this precision. Drills are already 5cm thick and just a very slight tilt results in several decimeter error. I doubt, that you have a sub dm accurracy which seems to be described in this paragraph.*

Your point about the precision of the data is certainly valid. However, the rope stretching, would cause positional differences of up to $0.5\,\mathrm{m}$, even if it some cases it was just a few centimetres, so all of them was interpolated. Additionally, location error due to tilted drilling will be minimized here because the ice was so relatively thin. We have rewritten this to hopefully make it clearer, the paragraph now ends as follows:
"*Finally, the rope stretching also means that the under-ice irradiance measurements were not performed in the same locations as the thickness drillings, with differences possibly up to $0.5\,\mathrm{m}$. To account for the rope stretching, ice thickness, pond depth and freeboard were interpolated linearly to the estimated locations of the irradiance measurements (fig. 4), though there remains some uncertainty in the exact locations of both the irradiance measurements and the thickness measurements.*"

*p.10 ll. 12-14: You describe the difference in sampling times, but you do not discuss it or even state that you dont think it affects your data.*

It was implicitly mentioned in lines 13–14, but we have rewritten to:

"*As the surface may have changed somewhat in the interim, the average intensity obtained from the image may not exactly represent the conditions at the time of the radiometric measurements.*"

> *p.10. ll.15-16: These statements feel more like a method description rather than a discussion. I suggest to move the remnants of this paragraph to the end of the discussion and present important discussion first.*

The sentence in question has been removed.

> *p.10. l.17. ff: The reason and purpose of this paragraph does not become evident. It is certainly valuable, but needs more motivation and explanation.*

We have updated this paragraph with a short motivation for the figure and also rewritten the discussion of the figure in a way we think makes its relevance clearer.

> *p.11 l.10: Obviously AccuRT can not account for anisotropic radiative transfer, however anisotropy is crucial to edge-effects at pond edges, so it should be mentioned also along the discussion of spatial scales (above general comment).*

The following has been added to the end of section 4.3:

"*Further, the vertical orientation of brine inclusions commonly seen in sea ice, appears to cause anisotropic scattering in the ice (Trodahl et al., 1989; Katlein et al., 2014), which AccuRT cannot account for. This issue may be important for determining the exact radius over which edge effects are important.*"

> *p.12 l. 12ff: This does not account for small spatial scales (see above). Furthermore this equation is written for transects only, but it would improve the usability of this paper if you suggest a form for aereal estimates.*

The paragraph in question has been extended with the following:

"*This approach can easily be applied to area fractions of surface types by replacing the lengths by areas.*"

> *Figure 8: All lables are missing. What is the relation between PAR and BB and at which depth can we use a conversion constant between them.*

A conversion constant between these two quantities and any relation between them will be strongly dependent on the properties of the ice and on the incoming spectrum. Clouds will reduce the amount of incident light outside the PAR range, decreasing the depth at which the two become nearly a constant multiple of each other. Increased scattering near the surface would allow for more interactions within a shallower depth, reducing the depth at which most non-PAR light has been absorbed. Overall, this seems like a complicated topic that cannot be properly addressed as a side note here.

> *p.15 l. 6-7: Your manuscript lacks a comparison of measured fluxes to existing datasets.*

Section 4.1 has been extended to include this.

> *p.15 l.14: I suppose it is also very much supporting the works of Arndt & Nicolaus (2014,Cryosphere) and Nicolaus (2012, GRL).*

Yes, and they have been cited.

> *p.15 l.15: This approach is not new. It has been demonstrated in Katlein et al. 2015 and also in some works from the campaign at hand (Divine et al.)*

Good point, rephrased to
   *"Obtaining information about ponded ice from aerial images as described above shows potential for this particular type of study, and has also been applied successfully in other studies (Divine et al., 2015; Katlein et al., 2015)."*

> *p.15 l.26: Pleas provide an accurate pointer (doi) for the associated dataset. A general pointer to a search engine is not enough.*

As the text stated, the dataset had not yet been published at the time of submission, so the DOI was not available. It has now been added.

[revised manuscript text omitted]

where $E_{\downarrow,0}$ the total scalar irradiance $E_0 = E_{0\uparrow} + E_{0\downarrow}$. From (1) and $E_{\uparrow,0}$ are the downward and upward scalar irradiance , respectively, obtained from model output along with $a$. (2), we use the model output to calculate local spectral heating rates as

10    $$H_\lambda = a(|E_{0\uparrow}| + |E_{0\downarrow}|). \tag{3}$$

Bulk transmittance over the range of wavelengths from $\lambda_1$ to $\lambda_2$ is defined as

$$T_{\text{bulk}} = \frac{\int_{\lambda_1}^{\lambda_2} T(\lambda) E_{\text{in}}(\lambda)\, \mathrm{d}\lambda}{\int_{\lambda_1}^{\lambda_2} E_{\text{in}}(\lambda)\, \mathrm{d}\lambda} = \frac{\int_{\lambda_1}^{\lambda_2} E_{\text{tra}}(\lambda)\, \mathrm{d}\lambda}{\int_{\lambda_1}^{\lambda_2} E_{\text{in}}(\lambda)\, \mathrm{d}\lambda} \tag{4}$$

where $T$ is spectral transmittance, $E_{\text{in}}$ is incident total irradiance (downward diffuse plus direct beam), and $E_{\text{tra}}$ is transmitted total irradiance. Unless otherwise specified, the wavelength range over which the bulk transmittance is calculated is 400–

15    900 nm. Bulk albedo is calculated similarly, but with reflected irradiance instead of transmitted irradiance in the numerator.

**2.4  Model setup**

Irradiance was calculated every 2 nm, and the resulting spectra smoothed with a Gaussian filter, to approximate the  spectral resolution of the RAMSES sensors.

Model parameters were chosen through an iterative process to obtain a good correspondence with the measured data.

20    In the radiative transfer model, the sea ice under ponds is represented by two layers (fig. 2). The upper layer has a thickness of 10 cm, with an air volume fraction that depends on the  average pixel intensity of the aerial photo (see below), with higher air volume fractions for brighter areas. Brighter areas appeared to be caused by a layer containing a larger amount of air, but the thickness of this layer in the simulations (10 cm) was chosen arbitrarily. The lower layer extends from 10 cm below the pond to the measured ice thickness, and has a fixed air volume fraction of 0.1%. Bare ice is also represented with

25    two layers, but here the thickness of the upper layer, corresponding to the surface scattering layer (SSL), is determined by the measured freeboard. The upper layer, which consisted of granular ice, was simulated as large-grained snow, i.e. ice spheres in air, with a radius of 2.5 mm. All ice, with the exception of the SSL, was simulated with a 20% brine volume fraction. The effective radius of air bubbles in all simulations was set to 0.25 mm, which is a bit higher than the upper bound for the air

[Figure]

**Figure 2.** Sketch of layers in AccuRT. Values for the effective scattering coefficient $b$ $b_{\text{eff}} = b(1-g)$ are averages over the visible range.

inclusions found by Light et al. (2003), though it should be noted that was for ice at a temperature of $-15°$C, whereas the ice in our study was warmer, around $-1°$C. For brine, the effective radius of inclusions (1.5 mm) was taken to be constant for all ice layers in the simulations.

Values of the effective scattering coefficient, $b_{\text{eff}} = b(1-g)$, are shown in fig. 2. Light et al. (2015) used $b_{\text{eff}}$ of $102\,\text{m}^{-1}$, $15\,\text{m}^{-1}$ and $\approx 1\,\text{m}^{-1}$ in the SSL, drained layer and interior ice, respectively. Hence, our SSL has less scattering than that of Light et al. (2015), but we did not include a drained layer. On the other hand, our interior ice has higher scattering. For a review of $b_{\text{eff}}$ used in various publications, see table 2 in Petrich et al. (2012).

Ponds are represented by adding a layer of pure sea water on top of the sea ice, with the thickness of the water layer equaling the measured pond depth. Water depth in the region where the measurements were taken is around $3\,\text{km}$, and that depth is used as the total water depth in the radiative transfer simulations. The sea water itself was pure sea water with the addition of absorption and scattering measured during the Norwegian Young Sea Ice Cruise in 2015 ("pre-bloom" case from Taskjelle et al., 2017a). The solar zenith angle was set corresponding to the time and approximate location of measurements, with values between $64.5°$ and $66.3°$.

The number of streams used in the upper slab, i.e. the atmosphere, was set to $N_{\text{U}} = 32$. The number of streams in the lower slab (ice and ocean) is set automatically by AccuRT to $N_{\text{L}} = N_{\text{U}} \cdot m_{\text{max}}^2$, where $m_{\text{max}}$ is the maximum refractive index of the lower slab in the simulated wavelength range, which gives $N_{\text{L}} = 57$ for ponded ice and $61$ for bare ice. These high numbers of streams were needed to avoid a numerical artifact in the calculation of albedo over the highly scattering bare ice.

Varying cloud cover was present on all days, causing large variations in incident irradiance. Integrated from $400$nm to $900$nm, values ranged from $84\,\text{W}\,\text{m}^{-2}$ to $266\,\text{W}\,\text{m}^{-2}$. In all simulations clouds are represented by a $0.5\,\text{km}$ thick layer of water droplets, with a base height of $2\,\text{km}$, and an effective droplet radius of $10\,\mu\text{m}$. The volume fraction of cloud particles

**Table 1.** Measurement summary for all measurements along the transects. $\bar{h}_{\mathrm{p}}$ is mean pond depth, $\bar{h}_{\mathrm{i}}$ is mean ice thickness, $\bar{h}_{\mathrm{fb}}$ is mean freeboard, and $\bar{T}$ is mean bulk transmittance [eq. (4)] in the range $400\,\mathrm{nm}$ to $900\,\mathrm{nm}$. Numbers in parenthesis indicate one standard deviation. $N_{\mathrm{p}}/N_{\mathrm{b}}$ is the number of spectra collected along the transect under ponded and bare ice, respectively. Positions are approximate.

| Date | Position | $N_{\mathrm{p}}/N_{\mathrm{b}}$ | Ponded ice | | | Bare ice | | |
| --- | --- | --- | --- | --- | --- | --- | --- | --- |
| | | | $\bar{h}_{\mathrm{p}}$ [m] | $\bar{h}_{\mathrm{i}}$ [m] | $\bar{T}$ | $\bar{h}_{\mathrm{fb}}$ [m] | $\bar{h}_{\mathrm{
[revised manuscript text omitted]